# Analysis of Rainfall Time Series with Application to Calculation of Return Periods

Ramón Egea Pérez [1], Mónica Cortés-Molina [2],* and Francisco J. Navarro-González [2]

1    Department of Planification et Travaux, Municipal Water and Sanitation Company, 30008 Murcia, Spain; regea@hotmail.es
2    Department of Applied Mathematics, University of Alicante, 03690 Alicante, Spain; francisco.navarro@ua.es
*    Correspondence: monica.cortes@ua.es

**Abstract:** This paper presents a study of the characteristics of rainfall in a typical Mediterranean climate, characterized by infrequent and irregular rain in the territorial area and its intensity. One of the main components of this type of climate is short-duration and high-intensity rain events that cause a large amount of damage to property and human lives, seriously affecting the operation of infrastructure and the activity of society in general. The objective of this study was to design a methodology based on peak over threshold (POT) analysis. This methodology allows us to establish reference precipitation values and more approximate return periods in the absence of sufficiently extensive historical precipitation series. In addition, the frequency of these extreme events or return periods is established. The characteristics of the precipitation regime make direct analysis difficult. Thus, the functions of the probability distributions underlying the described phenomena are improved.

**Keywords:** return period; extreme rainfall; Mediterranean climate





## 1. Introduction

On a regional or local scale, climate analysis and prediction techniques using numerical and statistical models are widely used. Episodes of isolated atmospheric depression at high levels (in Spanish, depresion aislada en niveles altos (DANA)), generated by the collision of a cold air mass at altitude with warm surface air, produce extreme precipitation events. These events have a return period, T, or estimated number of years that, on average, will equal or exceed a given value. Analyzing the frequency of these extreme events is essential for prediction and for the engineering design of infrastructure. Estimating such events can be done by analyzing precipitation time series associated with generally asymmetric or skewed distribution functions: Gumbel, generalized distribution of extreme values (GEV), SQRT-ETmax (statistical distribution function of extreme values used in the regional analysis of maximum hydrological values; see [1]), log-normal, Weibull, Fréchet, generalized Pareto distribution (GPD), and others [2]. In the case of the Mediterranean coast, there is a greater influence of the upper-level atmospheric circulation, which generates extreme episodes of precipitation. This phenomenon has been studied in [3,4], an approximation of which is made by means of decomposing singular values. This method isolates linear combinations of variables within two fields that tend to be linearly related to each other.

Rainfall during the month of March in the Iberian Peninsula underwent a sustained decrease of up to 70% between the 1950s and the 1990s. The most relevant rainfall occurs during September and October, with frequent episodes of torrential rainfall generated by high–level atmospheric depressions (DANA), or "cold drop", in the east of the peninsula [5–10].

Torrential precipitation is measured by the torrentiality index (I1/Id), defined as the relationship between hourly intensity, I1 (mm/h) and the corrected average daily precipitation intensity, Id (mm/h). Its value depends on the geographic area; on the Mediterranean

coast (east) of Spain, the value is 11, according to Standard 5.2.IC "Superficial Drainage", Order FOM/298/2016, as can be seen in Figure 1.

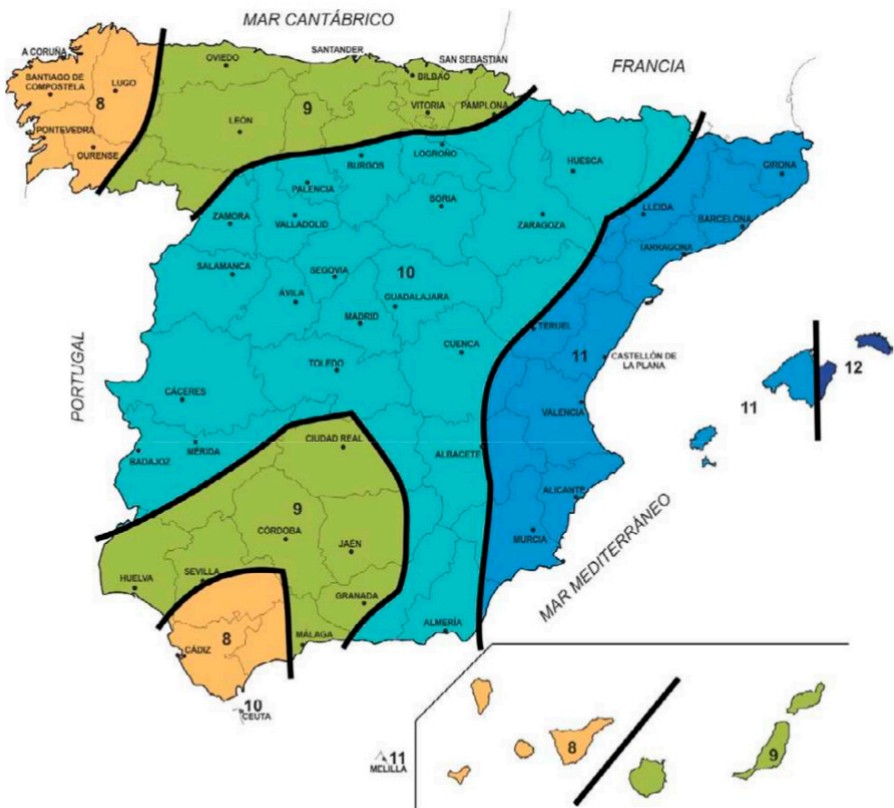

**Figure 1.** Map of torrentiality index (I1/Id) according to Standard 5.2. IC, Superficial Drainage, Order FOM/298/2016.

Changes in land use and the loss of vegetation cover as a consequence of anthropogenic activities as well as the progressive desertification of the soil, especially on the Mediterranean coast [11,12], are factors that contribute to the increasing effects of climate variability and thus the frequency of extreme precipitation events. There are two possible approaches to mitigate the resulting problems: optimize the use of resources directly or indirectly [13,14] or improve the predictive capability for the phenomena under consideration.

In the literature there are several approaches to studying hydrological extremes, such as analyzing the frequency with which the parameters of a given model vary as a function of time. The study of the concept of the return period and the risk of non-stationary hydrological phenomena to see how they can influence hydraulic structures is beginning to gain importance.

Many studies have focused on the variables that can influence future precipitation, such as evapotranspiration [15], air temperature, and atmospheric pressure, which affect climate change [16]. The impacts of climate change on extreme daily precipitation are remarkable and recognized in several studies [17–19].

Along the same line, studies on the surface temperature of the Mediterranean Sea can be found. This factor plays a very important role, as it acts as a source of humidity in Mediterranean cyclogenesis, affecting the conditions that produce torrential rainfall. These are explained by a mechanism that, by itself, allows the development of a potentially unstable mass over the Mediterranean Sea [20].

Other studies have investigated the factors that may influence precipitation related to greenhouse gases [21] and sulfate aerosols [22], although nothing can be concluded about their future influence on Mediterranean precipitation.

Consequently, a number of studies on changes in annual and seasonal precipitation amounts and their variability have emerged using different methods [23–26]. Related to variability is the problem of maximum or extreme values, with various mentions in the literature, such as an analysis of the distribution of maximum daily rain applying the SQRT-ETmax distribution model [1]. Other studies use generalized extreme value (GEV) distribution, considering series of very long annual maximums for the estimation and analysis of their suitability for the statistical treatment of real data, through the relationship between statistical parameters and climatological variables [27].

Studies have examined extreme events that occurred in the Segura Basin, Spain, and return periods considered as a SQRT-ETmax distribution function in the Mediterranean area (a new approach to pluviometric regionalization) [26,27]. To complete the analysis, the variables precipitation intensity, I (mm/h), estimated duration, D (minutes), and frequency, F (years), or probability of occurrence or the return period, T (years), of each extreme precipitation event are characterized and integrated in intensity–duration–frequency (IDF) curves. These IDF curves can be obtained by wavelet analysis, applying the different distribution functions described for extreme precipitation events. In this way, new return periods can be characterized [28–30].

## 2. Materials and Methods

### 2.1. Study of Probabilities of Extreme Events

Let us consider a set of values of a variable sampled at times $\{t_i / i \in \mathbb{N}\}$:

$$s_i = s(t)|_{t=t_i} \tag{1}$$

For simplicity, it is usually convenient to limit the sample times to instants homogeneously separated by a time interval given by h. Then, Equation (1) can be rewritten as:

$$s_i = s(t)|_{t=t_0 + h \cdot i}, \tag{2}$$

Examining the characteristics of signal $s(t)$ that generates the values of the data series $\{s_i / i \in \mathbb{N}\}$ is usually done from the point of view of estimating future values of the variable [31–41]. Another approach more related to the current research to decompose the signal into more basic components. A variety of techniques can be used for this goal: Fourier analysis, wavelet decomposition [42–45], and Hilbert–Huang transform (HHT) [46–51].

#### 2.1.1. Generalized Extreme Value (GEV) Distribution

Usually, statistics focuses on the properties of average values of stochastic processes leading to the central limit theorem. The main strength of this result is its validity irrespective of the particular distribution that generates the values. The study of extreme and rare values is a different branch of statistics with its own results, for example, the Fisher–Tippett theorem [52]. Before going into further detail, it is important to determine what is considered to be a rare event.

Extreme values theory (EVT) studies the properties of the distributions of extreme values of phenomena regardless of the underlying distribution of probabilities that generates these values. Let us consider a dataset of values obtained from an independent and identically distributed (iid) random distribution F and divide them into $m$ blocks containing $n$ values in each, $\left\{ X_1^{(i)} \ldots X_n^{(i)} \right\}_{i=1}^{m}$. Then, under certain criteria over F, the series of maximum values $\left( M^{(1)}, \ldots M^{(m)} \right)$, normalized as $\left( M^{(i)} - \mu \right)/\sigma$, distributes as the generalized extreme value (GEV) distribution with the cumulative distribution function parametrized by $\xi$:

$$H_\xi(x) = \begin{cases} exp\left(-(1 + \xi \cdot x)^{-1/\xi}\right) & \xi \neq 0 \\ exp(-e^{-x}) & \xi = 0 \end{cases} \tag{3}$$

Depending on the values of $\xi$, the specific distribution is called Weibull ($\xi < 0$), Gumbel ($\xi = 0$), or Fréchet ($\xi > 0$).

Analogous to the central theorem of the limit, for a large $n$, the following equation is satisfied:

$$P\left(M^{(i)} < x\right) = H_{\xi}\left(\frac{x - \mu}{\sigma}\right) \tag{4}$$

Although this approach to the problem of determining the characteristics of the distributions under study is often used, there is an alternative approach with better characteristics, as we will explain in the following section.

### 2.1.2. Peak over Threshold (POT) Pareto Distribution

Given that in each set $\left(X_1^{(i)} \ldots X_n^{(i)}\right)$ there exist values other than the maximum that are not being taken into account, the study of block maxima does not take advantage of all available information. For this reason, other approaches, such as peak over threshold (POT), are more efficient, and in practice this method has become more commonly used.

In the POT technique, the behavior of values above a reference level (threshold) are studied to obtain their distribution.

Let us consider a random variable following a distribution given by the function $f$, $X \sim f$. With $F$ denoting the cumulative distribution function, the distribution of values over a value $w$ is given by the expression:

$$F_w(x) = P(X - w \leq x | X > w) = \frac{F(x + w) - F(w)}{1 - F(w)} \tag{5}$$

If the normalized values of the maximums $M^{(i)}$ converge to the GEV distribution, for large values of $w$, the distribution $F_w(x)$ converges to the corresponding generalized Pareto distribution (GPD) with cdf given by the following:

$$G_{\xi,\beta}(x) = \begin{cases} 1 - (1 + \xi \cdot x / \beta)^{-1/\xi} & \xi \neq 0 \\ 1 - exp(-x/\beta) & \xi = 0 \end{cases} \tag{6}$$

The three cases given by the different values of $\xi$ (see Equation (3)) correspond to the ordinary Pareto, exponential, and Pareto II distributions.

### 2.2. Determination of Return Periods

Through the statistical analysis of precipitation, extreme precipitation events of an increasingly less exceptional nature can be linked, given their increasing frequency and temporal randomness over the last decades, so that the probability $P$ of the occurrence of such extreme events can be obtained. In hydrological analyses, the inverse of the probability of a certain event (return period, $T$) is often used:

$$T_i = \frac{1}{P_i}, \tag{7}$$

where $T_i$ is the return period of event $i$ and $P_i$ is the probability of occurrence of event $i$.

The return period $T_i$ or recurrence period of an extreme precipitation event is defined as the interval of years that, on average, a precipitation threshold value will be equaled or exceeded, which is equivalent to the frequency of occurrence $f$ of a given precipitation value:

$$f_i = \frac{1}{(1 - P_i)} \tag{8}$$

Although extreme precipitation events occur less frequently than ordinary precipitation, their occurrence is increasing progressively and with increasing randomness, which requires an exhaustive analysis of the distribution of the series of pluviometric values.

The probability of risk or occurrence of a given event $X$ at least once during a period of time $n$ is defined by

$$P(X \geq x_T) = 1 - \left(1 - \frac{1}{T}\right)^n \tag{9}$$

where $n$ is the number of years, $P$ is the probability of occurrence of an event, and $T$ is the return period.

Or, $R = 1 - (1 - P)^n$, where $R$ is the risk that a given infrastructure will be affected by a given magnitude of an extreme event and $P = 1/T$ is the probability of occurrence of such an extreme event of magnitude equal to or greater than the extreme event, in time period $n$.

On the other hand, another expression for the return period, $T = 1/\left(1 - e^{\frac{ln(1-R)}{n}}\right)$, is applicable in the case of risk analysis when the risk $R$ of an event occurring with probability $P = 1/T$ is known, as well as the number of years.

The probability of exceeding $k$ times a given rainfall in return period $T$ is given by a Poisson distribution:

$$P(k, t) = \frac{1}{k!}\left(\frac{t}{T}\right)^k e^{-\frac{t}{T}} \tag{10}$$

where $k$ is the number of occurrences of an event, $T$ is the return period (years), and $t$ is elapsed time (years).

In the hydrological analysis of extreme precipitation events, the following factors and proxy variables should be considered when analyzing the return period, T:

- Modification of hydrological channels;
- Variation of flows;
- River basin typology;
- Variation in land use in the area potentially affected by flooding;
- Infrastructure located in the analysis area;
- Climatological variability (effects of climate change);
- Orography of the study area;
- Geological characterization of the terrain in the analysis area;
- Links with other hydrological basins or nearby environment.

In summary, analyzing the return period and obtaining the IDF curves for each return period considering 5, 10, 25, 50, 100, 500, 1000, etc., years, and their linkage and adjustment with a certain distribution function of the series of pluviometric data analyzed allows characterization of the precipitation that has occurred and its possible potential effects, and forecasting of its temporal occurrence.

### 2.3. Numerical Model of Rainfall Time Series

The main problem when analyzing temporal series of meteorological data, especially rainfall registers, is the low proportion of an effective amount of data with values greater than zero. This can be a major problem in the case of regions and areas with a low rainfall profile.

Comparing data corresponding to different dates is then particularly complicated, since the values obtained in many cases do not correspond to actual occurrences of the phenomenon being studied.

To avoid this problem, one can consider that rainfall records, $\{x_i\}_{i=1}^N$, are generated by a process that only expresses itself effectively depending on a probability, $\pi(t)$, that varies throughout the year. At each instant, regardless of this realization, it is assumed that there is a "potential" distribution of precipitation given by $b(t)$. Additionally, some heteroscedastic ($\varepsilon_{het}$) and homoscedastic ($\varepsilon_{hom}$) perturbations can act over the average values:

$$s(t) = \pi(t) \cdot [b(t) \cdot (1 + \varepsilon_{het}(t)) + \varepsilon_{hom}(t)] \tag{11}$$

The nature of these perturbations includes seasonal effects and others caused by other frequency components.

The aim of this research was therefore to try to isolate the distribution, $b(t)$, and study the behavior of the extreme values of its distribution. In order to achieve this, the following steps were followed:

1.  Transform the original dataset, including only those days when $\pi(t) \geq 0$: there are many observations with zero value, which generates a distortion when calculating the mean values to compare non-zero data;
2.  Calculate monthly averages: In order to estimate the global trend (see next section), it is necessary to minimize the variance of the data considered. One way is to consider monthly groupings of the data, since this represents a period long enough for several rainfall episodes to appear and short enough so that the possible trend does not affect the calculated averages too much;
3.  Eliminate global trends: from the averages grouped by month calculated in the previous step, a linear regression is performed to obtain an overall trend;
4.  Remove seasonal effects of the time series from the median values of the monthly data grouped in step 2: once the medians (which are less sensitive than means to extreme values) of the monthly values are calculated, the difference between these values and the medians are used to study the seasonal components;
5.  Once the data are treated in steps 1–4, use the extreme values of distribution $b(t)$ to determine the characteristics of the distribution using the TOP approach: Steps 1–3 attempt to eliminate the main random values appearing in Expression (11). Once the global and seasonal components of the original signal are discounted, we can make the approximation $s(t) \approx b(t)$. The analysis of extremes is performed on this modified data;
6.  From the distribution obtained in step 5, calculate the return periods for different extreme values: once the distribution parameter estimates are obtained by POT, they can be used for return period forecasting;

Monthly averages are used to study global trends and seasonal averages are used to eliminate variability in the daily data. Once these trends are obtained, their effect is deduced from the daily dataset obtained in step 1.

## 3. Results

### 3.1. Introduction to the Rainfall Series of Ciudad Jardin Dataset

This dataset was previously used in [53]. The study area corresponds to daily precipitation values provided for the National Meteorological Agency (AEMET) for each day between 1 September 1938 and 31 January 2017 at a weather station located in the north of the city of Alicante, in southeastern Spain, as seen in Figure 2.

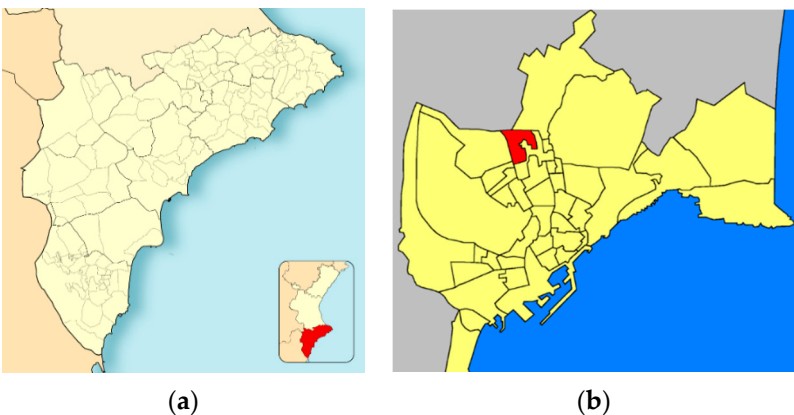

(**a**)    (**b**)

**Figure 2.** Location of weather station: (**a**) situation of city of Alicante in the country; (**b**) situation of Ciudad Jardin neighborhood in Alicante [54].

The data series is composed of 28,631 daily observations of precipitation between 1 September 1938 and 31 January 2017. Of these, 4720 records have values greater than zero and were selected to create the effective rainfall dataset. A summary of the full and effective datasets can be seen in Table 1.

**Table 1.** Summary of Ciudad Jardin dataset (full and effective).

|  | Date | Rainfall (L/24 h) | Effective Rainfall (L/24 h) |
|---|---|---|---|
| Minimum | 1 September 1938 | 0 | 0.1 |
| Maximum | 31 January 2017 | 270.2 [1] | 270.2 [1] |
| Median | - | 0 | 1.7 |
| Average | - | 0.8922 | 5.4694 |
| Std. Dev. | - | 4.9678 | 11.3687 |

[1] On 30 September 1997.

Figure 3 shows histograms of the $log(1 + rainfall)$ for the full and effective datasets, and Figure 4 presents the plot of the dataset. Obviously, the imbalance between the number of observations in the first case makes obtaining accurate results complicated. For this reason, only records with effective rainfall were considered in the modeling.

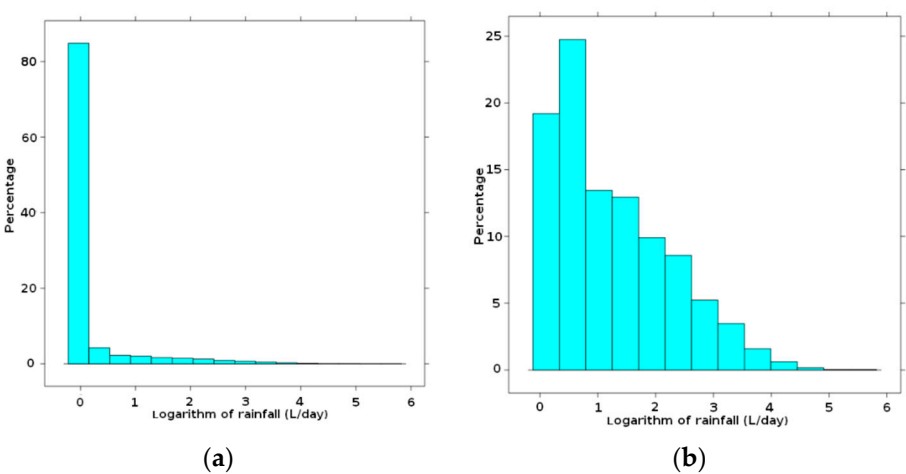

(**a**)　　　　　　　　　　　　　　　　　　(**b**)

**Figure 3.** Histogram of rainfall logarithms: (**a**) original full dataset; (**b**) effective dataset.

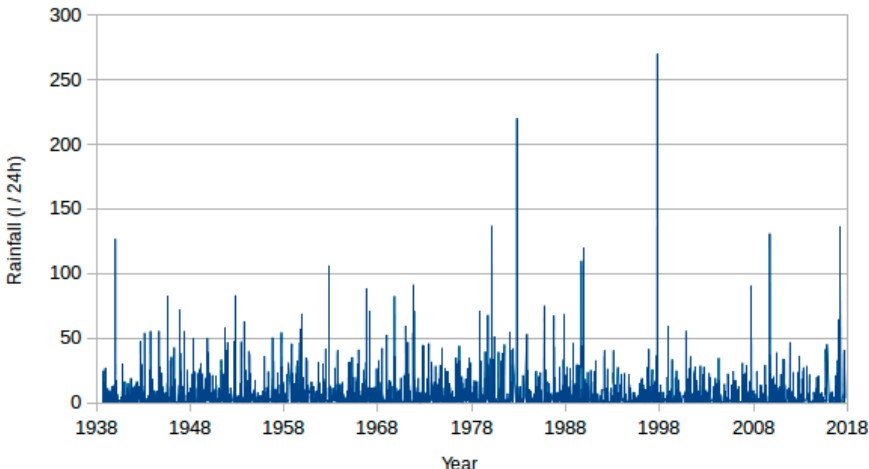

**Figure 4.** Effective rainfall data series.

A factor associated with the previous one to take into account is the number of consecutive days with precipitation (NCD). The corresponding summary is shown in Table 2.

**Table 2.** Summary of number of consecutive days with precipitation (NCD).

|  | NCD |
| --- | --- |
| Maximum | 10 [1] |
| Average | 1.7115804807 |
| Std. Dev. | 1.0895069514 |

[1] On 29 February 1948.

Given the characteristics of the rainfall regime in the studied zone, there were no long periods with continued precipitation, so correlation effects can be ignored. In addition, extreme events are not usually clustered in consecutive rainfall episodes.

*3.2. Filtering of Global Trends and Seasonal Components*

In order to study the existence of global and seasonal trends, the data were grouped into monthly and yearly records to reduce the inherent variance present in daily values. So, annual averages were used to study global trends and monthly grouped averages to account for seasonal relationships.

Given the results presented in the previous point, the average grouped by month was used to study the trends and main frequencies in the data series. This is to limit the variance in the data corresponding to daily rainfall using average rainfall as its representative magnitude (calculated over effective days with rain not past the 30/31 days of each month).

Once these components were calculated, their effect could be eliminated from the daily data series and the corresponding study of maximum components was carried out.

The data corresponding to October 1982, September 1985, September 1997, and March 2017 were not considered because they were influenced by the presence of extremely abnormal episodes; instead, they were considered as outliers in order to calculate the global trend of the series. Precipitation in these months corresponded to extremely important torrential rainfall episodes concentrated within a maximum interval of 3 days (see Table 2), which are precisely the type that motivated the present study. The magnitude of these episodes can be seen in Figure 5, with values for monthly average over 35 L/24 h.

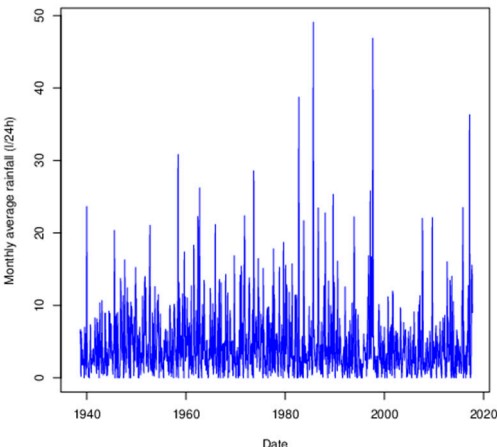

**Figure 5.** Evolution of monthly rainfall average.

3.2.1. General Trend

Once these values are removed, the data show a slight downward trend, as indicated by the regression line, represented in Figure 6:

$$rain = \alpha + \beta \cdot (year - 1938) \tag{12}$$

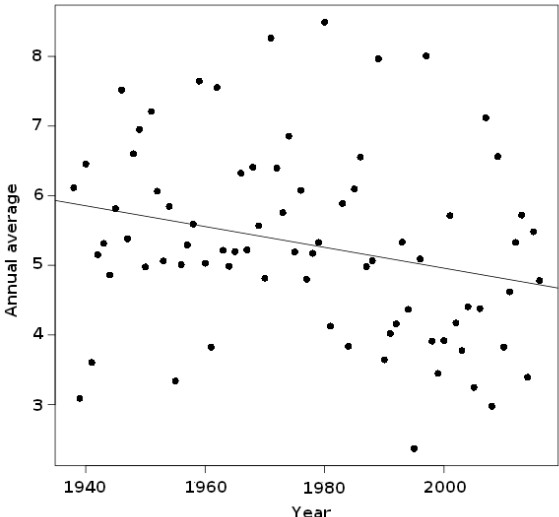

**Figure 6.** Evolution of annual rainfall average and linear regression.

The results of yearly average regression are shown in Table 3.

**Table 3.** Summary of linear regression from average monthly grouped data.

|  | Estimate | t Value Pr(>|t|) [1] |
|---|---|---|
| $\alpha$ | 34.826622 | 0.00747 ** |
| $\beta$ | −0.014934 | 0.02245 * |

[1] Level of significance of parameters (* = 0.90, ** = 0.95).

### 3.2.2. Seasonal Analysis

Following the steps introduced in Section 2.3, the next step is to investigate the periodic components of the signal without trend. So, the monthly averages were modified to delete the trend, followed by Hilbert–Huang transform analysis, as introduced in Section 2.1. The IMF result of the empirical mode decomposition (EMD) can be seen in Figure 7.

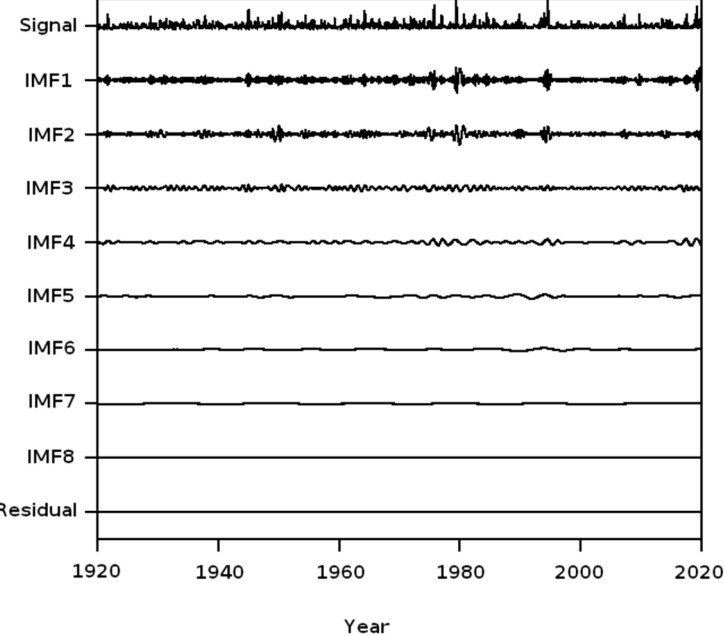

**Figure 7.** IMF components of empirical mode decomposition of trend-free monthly average rainfall signal.

The time–frequency representation for the principal IMFs (2, 3, 4, 5, and 6) can be seen in Figure 8.

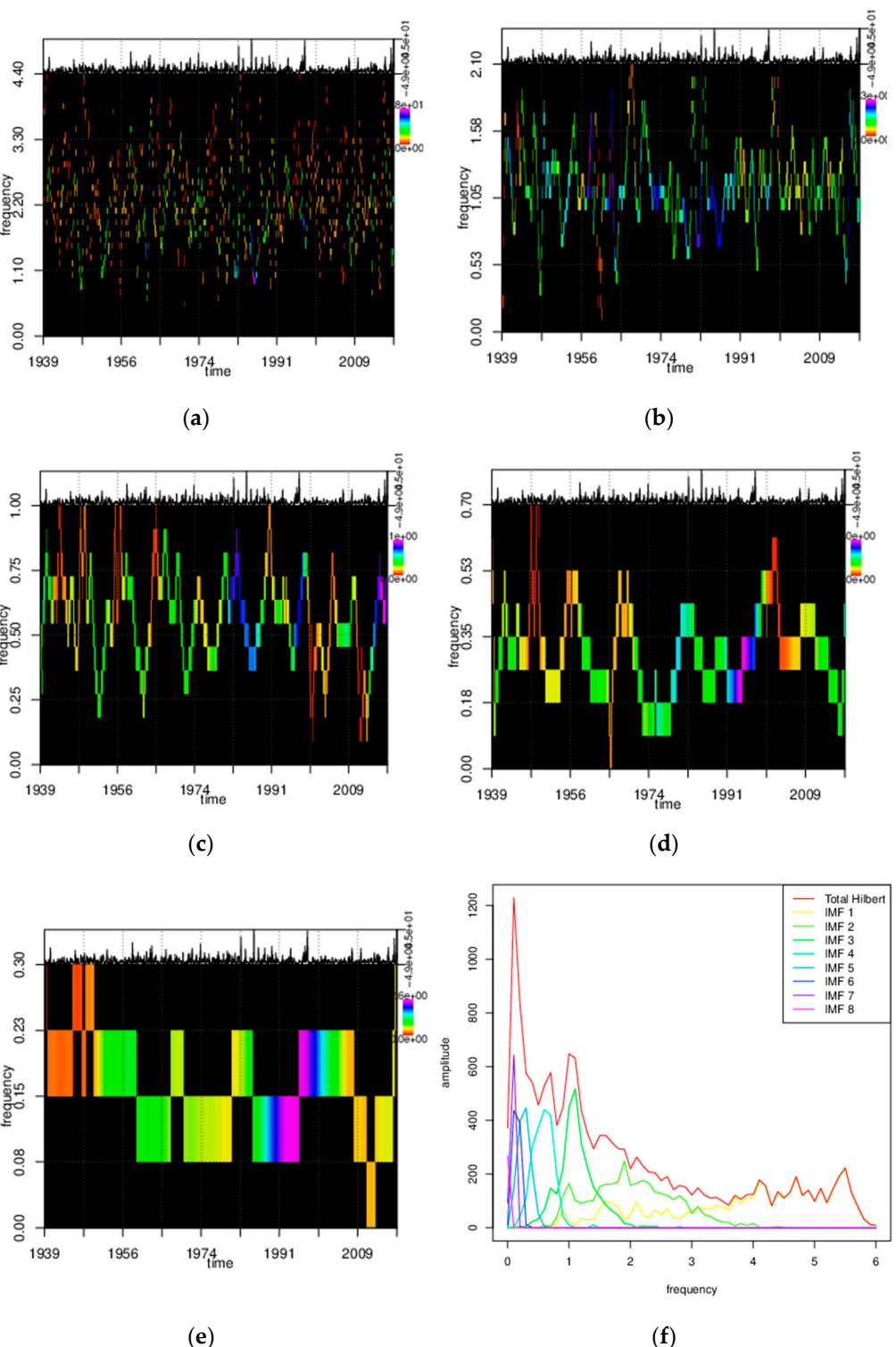

**Figure 8.** Spectrograms of main IMF components: (**a**) IMF2, (**b**) IMF3, (**c**) IMF4, (**d**) IMF5, and (**e**) IMF6; and (**f**) Hilbert periodogram with individual IMFs.

From the results of the Hilbert–Huang analysis, the most important periodic component corresponds to the annual period captured by IMF3 (Figure 8b). There are also other factors, corresponding to 6 months (IMF4, Figure 8c), 4 months (IMF5, Figure 8d), and

around 2 months (IMF6, Figure 8e), but their amplitudes are small except in the period 1991–1999, so they were not considered.

Additionally, comparing the evolution of the amplitude of each frequency over time in Figure 8a–e does not show a significant variation for the period of time between 1938 and 2017. To diminish this annual effect, a transform was made to the rainfall variable, subtracting the monthly median, which is less sensible to extremes than the mean:

$$rain[date] = rain[date] - median[month] \qquad (13)$$

### 3.3. Fitting of Distribution of Extreme Value Probabilities
### 3.3.1. POT Analysis

The most important step in estimating the parameters of the extreme value distribution is selecting the threshold. This has usually been done by using graphical tools that represent the behavior of statistical variables and identifying their behavior over the plots. As introduced in Section 2.3, representative examples of this approach can be seen in the plots in Figure 9.

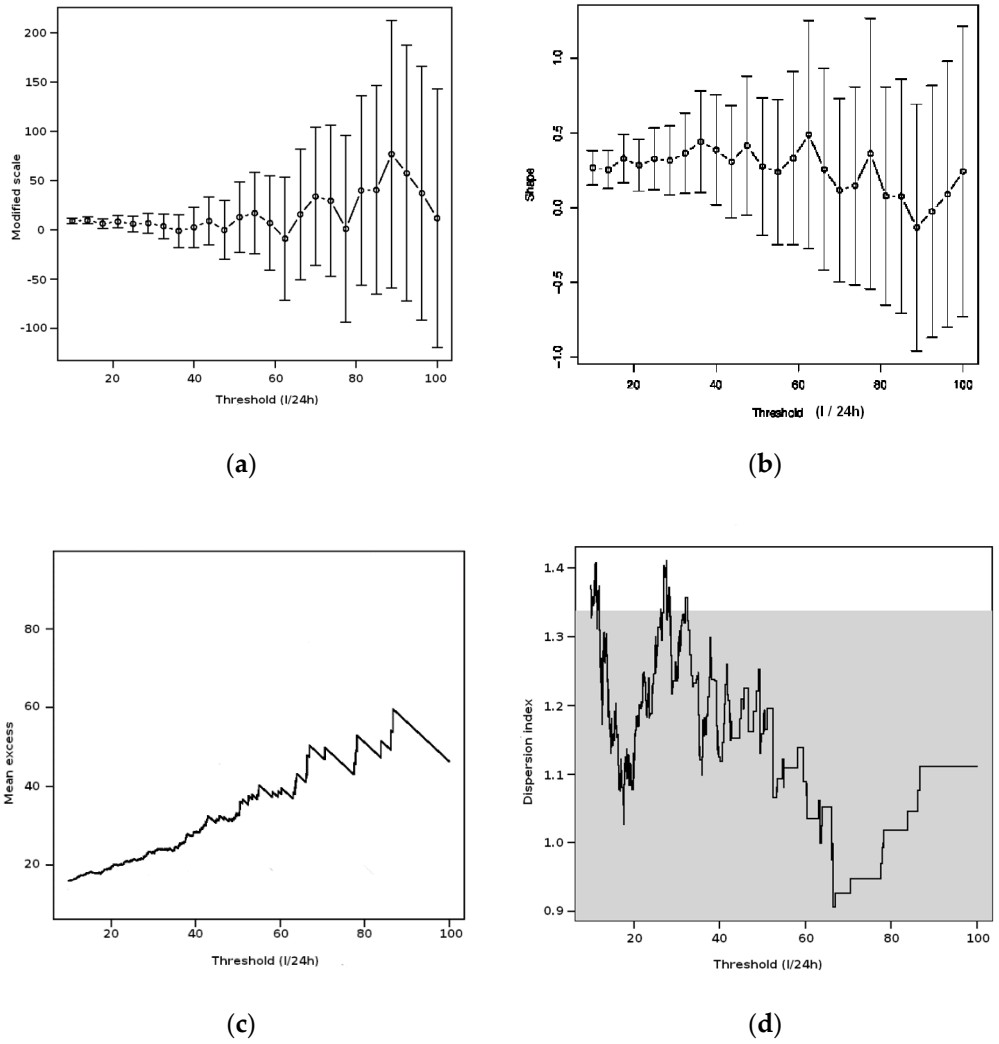

**Figure 9.** Graphic tools for threshold determination [55]: (**a**) TCP plot for scale parameter; (**b**) TCP plot for shape parameter; (**c**) mean residual life; (**d**) dispersion index plot.

The use of these tools, however, requires a certain degree of expertise. Even in that case, the result may not be evident at all. Fortunately, a new approach, based on the use of a hypothesis test, allows this step to be simplified.

First, let us remember that at this point the data used do not correspond with the original data with a domain given by $[0, \infty]$. Once the components are used as reference levels, the transformed data can present negative values corresponding to cases where the rainfall was under the expected value for the date. This will be used in the next step, given that the mean residual life plot in Figure 9c suggests looking for a threshold value not far from zero. Following the method introduced by Northrop [56], the result can be seen in Figure 10, which represents the $p$-values of the testing set $\xi_i = \ldots = \xi_m$ using the distribution $\chi^2_{m-i}$, against $u_i$ for the threshold set $(u_1, \ldots, u_m) = \{-6, -5, \ldots, 8\}$ (initially, the search range was between $-6$ and $60$, but for clarity in the results we show only the plot between $-6$ and $8$).

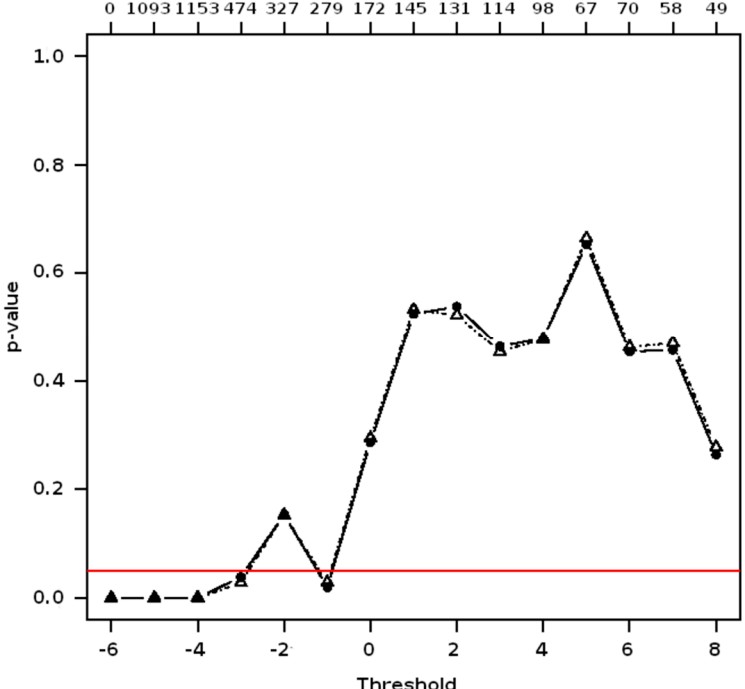

**Figure 10.** Threshold diagnostic plot (see [56]).

Using a $p$-value of 0.05 as the limit for the significance of the hypothesis test, the value of $-1$ can be taken as the threshold. Now it is possible to obtain estimations for parameters $\sigma$ and $\xi$ of Equation (6). Several estimators were used, and the results are shown in Table 4.

**Table 4.** Results for estimations of scale and shape parameters of generalized Pareto distribution.

| Estimator | Scale $\sigma$ | Shape $\xi$ | Scale Std. Error | Shape Std. Error |
|---|---|---|---|---|
| MLE | 6.2516 | 0.3927 | 0.26519 | 0.03597 |
| PWMU | 6.3134 | 0.3709 | 0.26627 | 0.03934 |
| PWMB | 6.3184 | 0.3704 | 0.26631 | 0.03928 |
| Pickands | 5.9410 | 0.5127 | – | – |
| MDPD | 6.0974 | 0.4245 | – | – |
| MPLE | 6.2677 | 0.3892 | 0.26486 | 0.03553 |

The values and standard errors are very similar, except for the estimators Pickands and MDPD. Using these results, one can calculate the average of the parameters weighted with the inverse of the corresponding standard errors to obtain a global estimation of $\sigma = 6.2562$ and $\xi = 0.3829$.

Additionally, some methods offer confidence intervals for these estimations, as can be seen in Table 5.

**Table 5.** Confidence intervals (0.95) for estimated scale and shape.

| Estimator | Inf. Scale $\sigma$ | Sup. Scale $\sigma$ | Inf. Shape $\zeta$ | Sup. Shape $\zeta$ |
|---|---|---|---|---|
| MLE | 5.731855 | 6.771363 | 0.3221609 | 0.4631557 |
| PWMU | 5.791555 | 6.835303 | 0.2938097 | 0.4480322 |
| PWMB | 5.796462 | 6.840393 | 0.2934333 | 0.4474126 |

To check the behavior of the model, we consider the estimator MLE and represent the quality of the model in Figure 11.

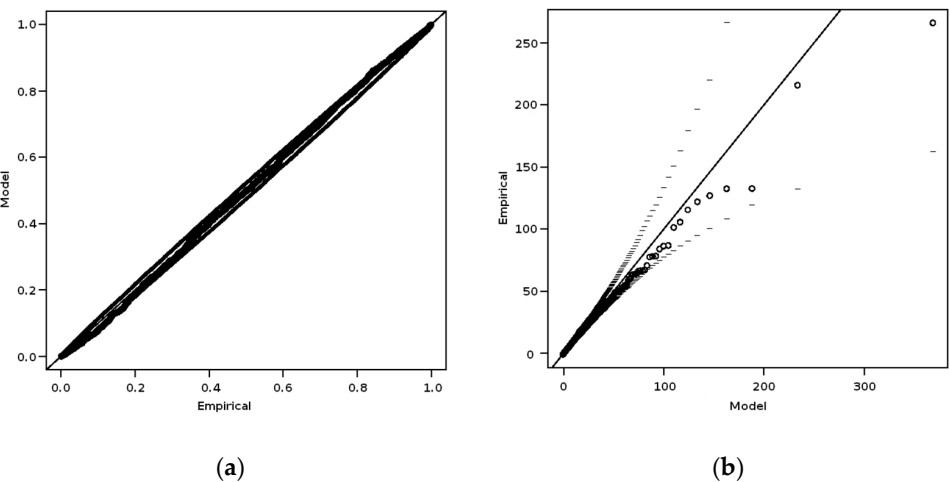

(a)                      (b)

**Figure 11.** Quality plots for estimator MLE: (**a**) empirical-model probabilities; (**b**) Q–Q plot.

### 3.3.2. Return Period

The number of rainy days per year can be averaged from the data series to obtain a result of 59.8974 with a standard deviation of 10.4354. Using these data, it is possible to represent the estimated period of return for 50, 60, and 70 events per year, as represented in Figures 12–14.

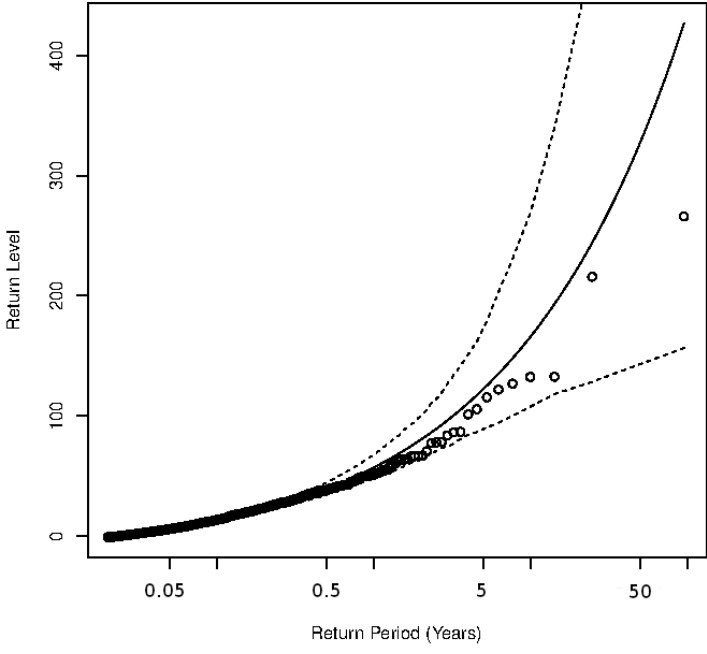

**Figure 12.** Return period for 50 annual events.

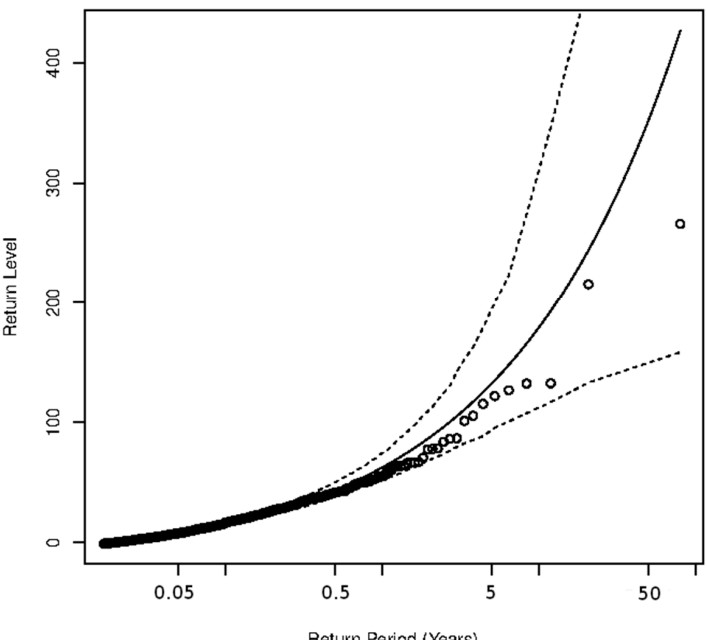

**Figure 13.** Return period for 60 annual events.

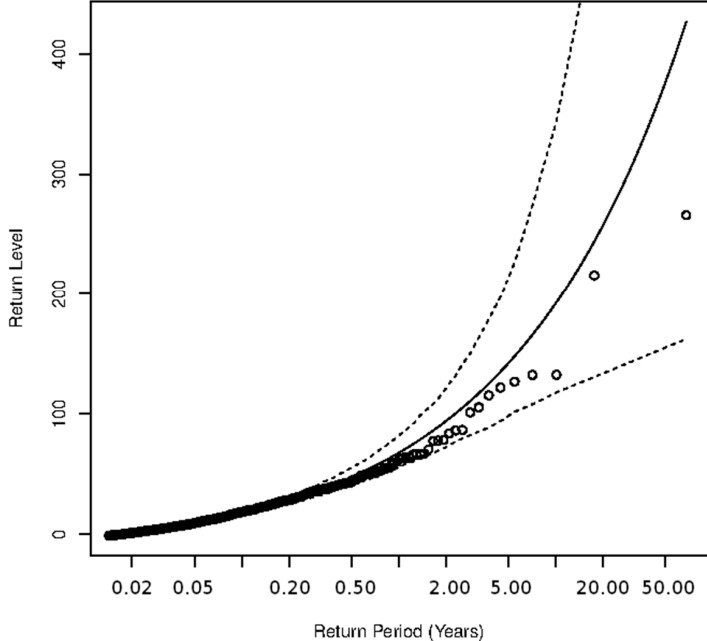

**Figure 14.** Return period for 70 annual events.

## 4. Discussion

Obtaining the return period T by means of the proposed analysis (Hilbert–Huang transform) is proposed as a recommendation for the analysis of extreme precipitation events.

Most of the studies we consulted and analyzed applied extreme event distributions relating mean maximum rainfall to return period T [2,17,25]. This implies that they did not take into account monthly, annual, and seasonal non-representative trends [28,30,31]. The proposed method more accurately considers the impact of extreme events on the ordinary rainfall regime (which may be affected by local climate change effects).

In contrast to other studies [1,5,6,9], in the analysis developed in this study, representative actual rainfall days (maximum ordinary rainfall and its annual average) are considered. In the analysis, disturbances generated by seasonal effects are isolated in order to study

their distribution, which is decisive in the behavior of extreme values [12,53,56]. It is also convenient to carry out the previous transformation of the data using monthly seasonal averages grouped together to study the possible global annual trend.

Finally, the use of the Hilbert–Huang transform allows the detection of periodic components [46], their period, and their importance in a clear and efficient way. A POT analysis was performed to obtain the parameters of the distribution of extreme events.

This research facilitates the identification of trends in the evolution and distribution of extreme precipitation values during extreme episodes [22,24], thus fixes the probable return period associated with rainfall of a given intensity, an essential value for hydrological planning and the design of hydraulic infrastructure [8,57].

## 5. Conclusions

The analysis developed in this research allows return period adjustments to be used in hydrological planning, related to the frequency of extreme episodes of precipitation used in the definition of intensities, and the frequency of extreme rain events, which are increasingly common and intense due to climate change.

The characteristics of the Mediterranean climate in the area under study, where generally infrequent rainfall of short duration and low magnitude alternates with torrential episodes, make the statistical study of these extreme events extremely complex. The biases introduced by the imbalance between the two regimes distort the results of the return period estimates.

The methodology proposed in this research does not make any assumptions about the probability distribution of rainfall values. Instead, it is based on the general theory of extreme values and assumes that the observed values represent an underlying process that manifests itself only on occasions when precipitation (potential rainfall) occurs.

To avoid the effects of the excess frequency of the zero value, the data were filtered so that only those recording actual precipitation values were considered. The general trends and seasonality effects were then extracted by performing linear regression and decomposition of the instantaneous frequencies of the signal grouped by months to reduce the statistical variability of the data. In order to make better use of the data, which are by their nature scarce, a POT analysis is proposed instead of other approaches frequently used in the literature. Once these values are estimated, they are used to generate return period curves.

The statistical analysis carried out provides relevant information along the same lines as most of the referenced research, related to estimating the distribution of frequency intervals of extreme precipitation events and, therefore, the return period as well as maximum daily precipitation intensity. It is noted that the higher the frequency of extreme precipitation events (higher probability of occurrence), the longer the return period. Likewise, there is a greater dispersion of precipitation intensity values (minimum and maximum) in line with an increased return period.

This research facilitates the identification of a trend in the evolution and distribution of extreme precipitation values during these extreme episodes and, thus, fixes the probable return period associated with a given rainfall intensity, an essential value for hydrological planning and the design of hydraulic infrastructures.

**Author Contributions:** Conceptualization, R.E.P. and F.J.N.-G.; methodology, F.J.N.-G.; software, M.C.-M.; validation, R.E.P.; formal analysis, F.J.N.-G.; investigation, M.C.-M.; resources, R.E.P.; data curation, M.C.-M.; writing—original draft preparation, F.J.N.-G. and R.E.P.; writing—review and editing, M.C.-M.; visualization, F.J.N.-G.; supervision, R.E.P.; project administration, M.C.-M. All authors have read and agreed to the published version of the manuscript.

**Funding:** This research was funded by the Municipal Water and Sanitation Company (EMUASA) and the Espinagua Company.

**Institutional Review Board Statement:** Not applicable.

**Informed Consent Statement:** Not applicable.

**Data Availability Statement:** Restrictions apply to the availability of these data. Data was obtained from Javier Valdés Abellán and Miguel Ángel Pardo Picazo and are available with the permission of Javier Valdés Abellán and Miguel Ángel Pardo Picazo.

**Acknowledgments:** The authors thank Javier Valdés Abellán and Miguel Ángel Pardo Picazo for their help in providing the dataset on which this research is based.

**Conflicts of Interest:** The authors declare no conflict of interest. The funders had no role in the design of the study; in the collection, analyses, or interpretation of data; in the writing of the manuscript; or in the decision to publish the results.

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
