# Peer review of "Analysis of Rainfall Time Series with Application to Calculation of Return Periods"

_sustainability, doi:10.3390/su13148051_

Round 1

Reviewer 1 Report

The article presents the important issue of extreme rainfall in Alicante, East Spain. For this purpose, the authors use a number of correctly selected statistical methods. Moreover, they indicate the utilitarian aspects of the discussed issue, extremely important for the functioning of the economy of the region.

However, the article requires some corrections and additions.

First of all, the article should be verified in terms of language. This is especially true of the first part of the article, where sentences are too long, sentence equivalents are present, some words are clearly missing in some places, which makes fragments of the text difficult to read. The description of the statistical methods is much better and there are virtually no errors there.

The titles of the drawings should be improved, as they should, if possible, contain information answering the question: what, where and when, in what research period. In Figure 1, the Spanish title should be removed. In Figure 2, the explanation is in the title, and it should be in the figure. Figure 3 is entitled Water flow between elements… although it presents a) location of Alicante in Spain and b) situation of the city. Both maps should be supplemented with a legend.

In Table 3, the title should be completed with the name of the station and the research period, because in this type of analysis the obtained results are strongly related to the location of the station and cannot be directly related to other stations. There is also no information about what the asterisks for t value mean.
Lines 250-261: the description of the next research stages should be standardized, e.g. point 2 requires clarification.
Line 228 contains Spanish text.
Figures 3 and 5 lack a description of the horizontal and vertical axes.

The main content-related remark concerns the lack of a discussion in the article understood as presenting the obtained results in the context of the research results of other authors, which would strengthen the thesis about the validity of the proposed methodology for the analysis of extreme precipitation. The content of the Discussion chapter should be moved to the next chapter of Conclusion.

Reviewer 2 Report

This is a paper dealing with the probability of extreme rainfall events to happen. This probability of occurrence is calculated through the concept of return periods. The analysis of rainfall extremes is quite beneficial for a variety of factors. However the manuscript faces several major and minor issues. The manuscript is difficult to follow, with several issues regarding the use of English. Below there are some remarks focusing on the introduction. However, an extensive proofreading is necessary for the whole manuscript. The most important issue however is that the manuscript seems to be written in a hurry. Several mistakes are found. There are mistakes in the figure caption numbers. There are spanish sentences left. It seems that an incomplete version has been submitted. The scientific part could be interesting. In order to perform a quality review, a more carefully structured and written version should be submitted.

Bellow are some remarks. However there are way more issue than what I have mentioned.

Introduction remarks:

Line 14: (society. in general) Remove the bullet.

Lines 14-20: Break this in more sentences.

Line 25: Change "are" to "is"

Line 26: What is DANA?

Lines 25-32: This sentence is huge and difficult to follow.

Line 34: Please rephrase "at altitude..."

Lines 35-36: "in which an approximation is made by means of the decomposition of singular values". Please start a new sentence explaining better what you mean.

Lines 37-43: Again. The whole paragraph consists of one sentence. Please break it into smaller sections. It is difficult to follow and confusing.

Lines 44-50: Too technical for the introduction. Please remove them.

Figure 1: Should be removed as well.

Lines 51-53: Irrelevant.

Line 82: What is the "SQRT-ETmax distribution model" ?

Lines 83-84: "General Distribution of Extreme Values (G.E.V.)". G.E.V. is the Generalized Extreme Value distribution! Please change it accordingly.

Lines 87-88: Irrelevant. If you still want to have this information, please implement it in another paragraph.

Other remarks

In Material and Methods, in the section 2.1. Time Series, some of the information provided is not used and does not help the reader in better understanding the methodologies used here or the finding. Therefore, it is unnecessary.

The GEV description should be removed or minimized. There is a figure describing its PDF. At the same time no similar description is provided for POT, which is the one used in Extreme Value Analysis.

Line 228: A spanish sentence is here!!!

Figure 3 is not referred anywhere.

Figure 110 should be Figure 11.

Figure 110 (11) needs to be uploaded in better resolution. 

Author Response

This is a paper dealing with the probability of extreme rainfall events to happen. This probability of occurrence is calculated through the concept of return periods. The analysis of rainfall extremes is quite beneficial for a variety of factors. However the manuscript faces several major and minor issues. The manuscript is difficult to follow, with several issues regarding the use of English. Below there are some remarks focusing on the introduction. However, an extensive proofreading is necessary for the whole manuscript. The most important issue however is that the manuscript seems to be written in a hurry. Several mistakes are found. There are mistakes in the figure caption numbers. There are spanish sentences left. It seems that an incomplete version has been submitted. The scientific part could be interesting. In order to perform a quality review, a more carefully structured and written version should be submitted.

Bellow are some remarks. However there are way more issue than what I have mentioned.

Reviewer 3 Report

This paper investigated the characteristics of rainfall in a typical Mediterranean climate. The study used daily rainfall data from 09/01/1938 to 1/31/2017 to analyze the global and seasonal trends. In general, I think the paper does a good job in terms of serving the main goal which is to design a method based on the peaks over threshold (POT) analysis for estimating the rainfall return period.

However, a few issues need to be addressed before being published. And a thorough editorial review mush be conducted to fix all the typo and grammar errors. I’m pretty sure that I was not able to capture all the ones.

Specific Comments:

  1. Line 14: Delete period before “in general”.
  2. Line 15: “focuses” should be “focusing”.
  3. Line 26: What is the “DANA”? Can you spell out it?
  4. Line 267: Can you briefly introduce the dataset you’re using? Like is the rainfall from the gauge observation? If yes, has it been processed/interpolated to pixel values? What is the spatial resolution of the data?
  5. Line 267: The reference number is not correct. The website link in [60] shows the location map.
  6. Lines 304-307: Can you add the reference for considering these months are influenced by the abnormal episodes?
  7. Line 347: “allow” should be “allows”.
  8. Can you add the x-axis (threshold) unit for Figures 10 and 11?
  9. Line 350: Typo for Figure 11 instead of Figure 110.
  10. Line 359: Missing the period at the end of the sentence.
  11. Line 368: Missing the period at the end of the sentence.

Author Response

This paper investigated the characteristics of rainfall in a typical Mediterranean climate. The study used daily rainfall data from 09/01/1938 to 1/31/2017 to analyze the global and seasonal trends. In general, I think the paper does a good job in terms of serving the main goal which is to design a method based on the peaks over threshold (POT) analysis for estimating the rainfall return period.

However, a few issues need to be addressed before being published. And a thorough editorial review mush be conducted to fix all the typo and grammar errors. I’m pretty sure that I was not able to capture all the ones.

Specific Comments:

Round 2

Reviewer 2 Report

Despite the efforts of the authors, the manuscript faces several issues and cannot be considered for publication. First of all the authors should consider addressing a native english speaker as there is a need for extended proofreading. They should also review the manuscript as there are many typos. For example in the Keywords section "periods return" should be rephrased to "return periods". Also there are two sections named the same (4. Discussion and 5. Discussion).

This supports my comments in the first round of reviews that the most important issue is that the manuscript seems to be written in a hurry.

I would also suggest removing unnecessary information like 2.1 Time series as they add noise to the paper. Also 3. Results and 4. Discussion should be combined into one section as it is easier for the reader to follow. Presenting the results and discussing them on the same page is more productive.

Author Response

Attached revised document.